# Preconditioned Mesenchymal Stromal Cell-Derived Extracellular Vesicles (EVs) Counteract Inflammaging

**DOI:** 10.3390/cells11223695

**Published:** 2022-11-21

**Authors:** Cansu Gorgun, Chiara Africano, Maria Chiara Ciferri, Nadia Bertola, Daniele Reverberi, Rodolfo Quarto, Silvia Ravera, Roberta Tasso

**Affiliations:** 1Department of Experimental Medicine (DIMES), University of Genova, 16132 Genova, Italy; 2Department of Neuroscience, Rehabilitation, Ophthalmology, Genetics and Maternal Child Sciences (DINOGMI), University of Genova, 16132 Genova, Italy; 3U.O. Molecular Pathology, IRCCS Ospedale Policlinico San Martino, 16132 Genova, Italy; 4U.O. Cellular Oncology, IRCCS Ospedale Policlinico San Martino, 16132 Genova, Italy

**Keywords:** inflammaging, extracellular vesicles, macrophages, mesenchymal stromal cells

## Abstract

Inflammaging is one of the evolutionarily conserved mechanisms underlying aging and is defined as the long-term consequence of the chronic stimulation of the innate immune system. As macrophages are intimately involved in initiating and regulating the inflammatory process, their dysregulation plays major roles in inflammaging. The paracrine factors, and in particular extracellular vesicles (EVs), released by mesenchymal stromal cells (MSCs) retain immunoregulatory effects on innate and adaptive immune responses. In this paper, we demonstrate that EVs derived from MSCs preconditioned with hypoxia inflammatory cytokines exerted an anti-inflammatory role in the context of inflammaging. In this study, macrophages isolated from aged mice presented elevated pro-inflammatory factor levels already in basal conditions compared to the young counterpart, and this pre-activation status increased when cells were challenged with IFN-γ. EVs were able to attenuate the age-associated inflammation, inducing a decrease in the expression of TNF-α, iNOS, and the NADase CD38. Moreover, we demonstrate that EVs counteracted the mitochondrial dysfunction that affected the macrophages, reducing lipid peroxidation and hindering the age-associated impairment of mitochondrial complex I activity, oxygen consumption, and ATP synthesis. These results indicate that preconditioned MSC-derived EVs might be exploited as new anti-aging therapies in a variety of age-related diseases.

## 1. Introduction

Biological aging can be defined as a loss of tissue’s physiological function both at the cellular and molecular level [1]. Different hallmarks, including genomic instability, telomere shortening, epigenetic alterations, mitochondrial dysfunction, senescence, and altered intercellular communication characterize biological aging [2,3]. Since the 1990s, a series of experimental data have indicated that the progressive increase in the concentration of circulating pro-inflammatory molecules was one of the primary features of human aging [4,5,6]. The first evidence of this phenomenon dates back to a 1993 paper, reporting a higher production of pro-inflammatory cytokines by peripheral blood mononuclear cells from elderly subjects in comparison with young individuals [7]. It was later proven that, on average, aged individuals (>50 years old) present a 2- to 4-fold increase in serum levels of pro-inflammatory mediators (e.g., interleukin-6 and tumor necrosis factor-α) compared with younger people [8,9]. Therefore, the term “inflammaging” has been coined to indicate a sterile, low-grade and chronic inflammation that progressively increases with age [10]. Recent evidence has shown that, beyond the central position of the immune system, cellular senescence plays an important role in modulating the inflammatory status [11]. Indeed, senescent cells exhibit a remarkable ability to modify the surrounding environment by secreting a set of molecules, termed SASP (senescence-associated secretory phenotype) [12,13,14]. The SASP is a pro-inflammatory secretome primarily characterized by the strong upregulation of several families of soluble signalling factors, including interleukins, chemokines and growth factors, that are responsible for the establishment of a senescent microenvironment [15]. 

Macrophages are intimately involved in initiating and resolving inflammation, and their dysregulation with age is a major contributor to inflammaging [16]. During aging, a significant reduction in tissue nicotinamide adenine dinucleotide (NAD) occurs and pro-inflammatory macrophages express a NAD-consuming enzyme (CD38), whose expression is increased by the SASP released by the surrounding environment, resulting in a consequent age-related NAD decline at tissue level [17,18]. 

Many studies have shown that Mesenchymal Stromal Cells (MSCs) possess the ability to resolve inflammation and promote tissue repair in different models of inflammatory diseases, mainly due to their paracrine activity [19]. Indeed, according to an increasing amount of evidence, the therapeutic effects of transplanted MSCs are not always dependent on their engraftment at the site of injury, mostly as a result of what they release in the surrounding environment [20]. In addition to the release of a variety of soluble factors, a fundamental role has been attributed to the paracrine signalling performed by extracellular vesicles (EVs), a family of membrane-bound vesicles with heterogeneous contents, including genetic materials, proteins, lipids, and metabolites [21]. We have previously demonstrated that the in vitro preconditioning of human-MSCs with hypoxia and inflammatory stimuli (Hyp^INF^) was able to modulate the cell paracrine activity [22]. 

The present study aims to extend the hypothesis that preconditioned MSC-derived EVs can exert an anti-inflammatory role in the context of inflammaging, acting as mediators of the dynamic interplay between MSCs and macrophages. We observed that aged mice (78–104 weeks old) are characterized by a significantly higher number of bone marrow (BM)-nucleated cells than young mice (8–12 weeks old), and their BM-derived macrophages have a pro-inflammatory profile, already in basal conditions, which increases significantly following stimulation with IFN-γ, a typical activator of the M1 polarization. It is noteworthy that the aged macrophages were largely non-responsive toward LPS stimulation, and this is in part due to a metabolic dysregulation [23]. The aged macrophages displayed a lower and less efficient oxidative phosphorylation than the young cells, causing a reduction in the aerobic energy metabolism and an oxidative damage accumulation, as shown by the malondialdehyde content. However, the aged macrophages showed enhanced anaerobic glycolysis, suggesting an attempt to counteract the mitochondrial dysfunction. Moreover, ageing led to an accumulation of CD38-positive BM-macrophages that could be induced by the SASP factors present in their BM environment. Interestingly, EVs released by Hyp^INF^-MSCs were able to revert this profile, exerting an anti-inflammatory role on the aged macrophages. The TNF-α levels, as well as the percentage of iNOS-positive inflammatory macrophages, were significantly down-regulated in the EV-treated aged cells. More importantly, the treatment, for 24 h with 0.5 μg EVs, induced a significant decrease in the CD38-expressing aged cells. In addition, treatment with EVs improved the efficiency of oxidative metabolism and cellular energy status in aged macrophages, both in basal and upon M1 stimulation, reducing the oxidative damage. In addition, the young macrophages showed a similar trend, which was only statistically significant upon stimulation with LPS and IFN-γ. The EV-associated effect could depend on an indirect mechanism mediated by the transfer of miRNAs or organelles [24], but also on the respiratory activity of the EVs themselves [25,26].

Taken together, these data indicate a possible use of MSC-EVs to reduce the levels of inflammation and senescence. This could open up new perspectives for the therapeutic use of EVs in the context of tissue repair, suggesting the possibility of enhancing the endogenous reparative mechanisms by manipulating the dysregulated inflammatory system.

## 2. Materials and Methods

### 2.1. Mice 

C57Bl/6 (MHC H2b haplotype) mice were used. All mice were bred and maintained at the Animal Facility of IRCCS Ospedale Policlinico San Martino. All animal procedures were approved by the Local Ethical Committee and performed in accordance with the national current regulations regarding the protection of animals used for scientific purposes (D. Lgs. 4 Marzo 2014, n. 26, legislative transposition of Directive 2010/63/EU of the European Parliament and of the Council of 22 September 2010 on the protection of animals used for scientific purposes). 

### 2.2. Isolation and Culture of Human Adipose Tissue-Derived Mesenchymal Stromal Cells (hMSCs)

The hMSCs were isolated from subcutaneous adipose tissue obtained from liposuction after informed consensus by Regione Liguria Ethical Committee authorization (P.R. 23571), as previously described [22]. The adipose tissue was washed with cold phosphate-buffered saline (PBS) and digested with 0.1% type I collagenase (Gibco, Milan, Italy) in PBS at 37 °C for 60 min. The digested tissue was centrifuged at 500× *g* for 10 min and the stromal vascular fraction (SVF) pellet was washed twice in PBS. The SVF was plated at a density of 2 mL of initial tissue/54 cm^2^ surface area and cultured in Dulbecco’s D-MEM High Glucose (Biochrom GmbH, Berlin, Germany) supplemented with 10% fetal bovine serum (FBS) (Gibco, Milan, Italy), 2 mM of L-glutamine, 100 U/mL penicillin, 100 μg/mL streptomycin (Euroclone) and 1 ng/mL of fibroblast growth factor-2 (FGF-2) (Peprotech, Milan, Italy). The cultures were maintained at 37 °C, 5% CO_2_. Every 3 days the culture medium was changed, and cells expanded until 85% of confluence using 0.005% of trypsin-EDTA (Gibco, Milan, Italy). Only cells from P1 or P2 passages were used for extracellular vesicle separation.

### 2.3. Isolation and Culture of Mouse Bone Marrow-MSCs (mBMSCs) 

The mBMSCs were isolated from either 8–12 weeks old (young mBMSCs) (*n* = 3) or 78–104 weeks old (aged mBMSCs) (*n* = 3) C57Bl/6 mice by flushing the bone marrow from femurs and tibias. The bone marrow cells were rinsed twice in PBS and the nucleated cells were counted by Methyl Violet staining. The nucleated cells were cultured in Alpha-MEM Glutamax (Gibco, Milan, Italy) supplemented with 10% FBS (Gibco, Milan, Italy), 2 mM of L-glutamine, 100U/mL penicillin, 100 μg/mL streptomycin and 1 ng/mL of fibroblast growth factor-2 (FGF-2) (standard medium). The cells were maintained in an incubator at 37 °C, 5% CO_2_, changing the medium every 3 days. Only cells from P1 or P2 passages were used for further analysis and secretome isolation. Both young and aged mBMSCs cultured in standard medium were analyzed for the expression of the following markers, using monoclonal antibodies against CD44 (Clone IM7, BD Biosciences, Milano, Italy), H2Kb (Clone AF6-88.5, BD Biosciences), CD105 (Clone MJ7/18, Biolegend, San Diego, CA, USA), CD29 (Clone HMb1-1, eBioscience, Waltham, MA, USA), CD90.2 (Clone 30-H12, BD Biosciences), CD45 (Clone 30-F11, BD Biosciences), CD31 (Clone 390, eBioscience), CD11a (Clone 2D7, BD Biosciences) and CD34 (Clone 4H11, eBioscience).

### 2.4. Collection of hMSC and mBMSC Conditioned Medium (CM)

When either hMSCs or mBMSCs (P2 passage in culture) reached 80% of confluence, they were rinsed three times in PBS to remove any residue of FBS. The cells were then replenished with a serum-free (SF) medium (D-MEM for hMSCs or alpha-MEM for mBMSCs) supplemented with 2 mM of L-glutamine, 100 U/mL penicillin and 100 µg/mL streptomycin, without FBS, for 24 h. 

For the preconditioning of the hMSCs, the SF medium was discarded, and the cells were cultured in the SF medium supplemented with inflammatory stimuli (50 ng/mL Tumor Necrosis Factor-α (TNF-α) (PeproTech, London, UK) and 50 ng/mL Interleukin 1-α (IL1-α) (PeproTech, London, UK) under hypoxic condition (1% O_2_) (Hyp^INF^) for 24 h. 

Subsequently, the CM was collected from both cell sources and concentrated with Amicon-Ultra-15, 3 kDa centrifugal filter tubes (Merck, Darmstadt, Germany). In order to quantify the protein amount, a BCA Protein Assay Kit (Thermo Fisher Scientific, Waltham, MA, USA) was used. The CM samples were then stored at −80 °C and used in experiments as 50 μg.

### 2.5. Immunohistochemistry

Femurs from the aged and young C57Bl/6 mice were cleaned from adherent tissue and maintained in EDTA solution (0.5 M, pH: 7.2–7.4) on a shaker. When a complete decalcification process was reached, the femurs were washed three times in PBS and included in paraffin using a standard histological technique. Five-micrometer serial sections were cut and used for immunohistochemistry studies. To detect the presence of iNOS-positive cells, the sections were treated with an anti-iNOS antibody (ab15323) (Abcam, Cambridge, UK). A heat mediated antigen retrieval was performed in a microwave with sodium citrate buffer (10 mM Sodium citrate, 0.05% Tween 20, pH 6.0) and left at RT for 20 min. Non-specific sites were blocked with a solution of 10% NGS, 1% BSA (Sigma, Darmstadt, Germany) in TBS at RT for 15 min. The slides were incubated with the primary antibody diluted (1:100) in TBS-1% BSA overnight at 4 °C and then detected using a biotinylated goat anti-rabbit secondary antibody (Dako Cytomation, Milano, Italy). Negative controls with pre-immune serum were run in parallel. The images were taken by Leica DMi1 microscope (Leica Microsystems, Milano, Italy).

### 2.6. Extracellular Vesicle (EV) Separation and Characterization

The EVs were separated from CM derived from hMSCs cultured in Hyp^INF^ conditions by differential centrifugation using a Beckman Coulter ultracentrifuge (Optima L-90K). CM was subjected to initial centrifugations to remove cells/cell debris (300× *g* for 10 min) and apoptotic bodies (2000× *g* for 20 min). The resulting supernatant was transferred to the ultracentrifuge tubes (Beckman-Coulter, Brea, CA, USA) and subjected to a first centrifugation at 10,000× *g* (10K) for 40 min to obtain a pellet enriched of medium-sized EVs (mEVs). The supernatants were further centrifuged at 100,000× *g* (100K) for 120 min to obtain a pellet enriched of small-sized EVs (sEVs). The pellets were then washed in sterile PBS and centrifuged at 100K. The pellets from 10K and 100K were pooled together, in order to consider the set of all released vesicles. The cell-surface proteins expressed by EVs were quantified by the BCA Protein Assay Kit. SW28 and SW55Ti swinging bucket rotors (Beckman Coulter, Milano, Italy) had been used during the ultracentrifugation steps. The EV concentration and the size distribution were analyzed by the ZetaView^®^ TWIN-NTA PMX-220 (Particles Metrix GmbH, Ammersee, Germany) according to the manufacturer’s protocol. The reading parameters were set as 80% sensitivity and 100 shutter for the light scattering mode. The expression of the tetraspanin family members CD81 (BV421 Mouse Anti-Human CD81, BD Biosciences, 740079), CD63 (PE-Cy7 Mouse Anti-Human CD63, BD Biosciences, 561982) and CD9 (APC Mouse Anti-Human CD9, Biolegend, 312108) was analysed by non-conventional flow cytometry, as previously reported [27]. The separated EVs were analysed by western blot [3], using the following antibodies: anti-syntenin (1:1000 dilution, ab133267, Abcam), anti-flotillin-1 (1:10,000 dilution, ab41927, Abcam), anti-Grp94 (1:1000 dilution, ab238126, Abcam), and goat anti-rabbit/mouse secondary antibody (1:1500 dilution, Cell Signaling Technology, Danvers, MA, USA).

### 2.7. Macrophage Isolation and Culture

The macrophages (MΦ) were isolated from the bone marrow of either the young (8–12 weeks old, *n* = 40) (yMΦ) or the aged (78–104 weeks old, *n* = 40) (aMΦ) C57Bl/6 mice. The bone marrow cells were flushed out from the femurs and tibiae and collected in 15 mL tubes. The nucleated cells were rinsed twice in PBS, counted, and subsequently resuspended in IMDM medium (Aurogene) supplemented with of 10% FBS, 2 mM L-glutamine, 100U/mL penicillin and 100 µg/mL streptomycin, and Macrophage-Colony Stimulating Factor (M-CSF) (10 ng/mL) (Peprotech, London, UK) (complete medium), and plated at a density of 2 × 10^6^ cells/well in 6-well plates. The medium was changed after 3 days. The cells were used for downstream analysis after 7 days in culture. The concentration of TNF-α was measured using the commercially available kit (Duo Set, R&D systems, Minneapolis, MN, USA). Both the yMΦ and aMΦ cultured in complete medium were analysed for the expression of the following markers, using monoclonal antibodies against F4/80 (Clone BM8, eBioscience), CD11c (Clone N418, eBioscience), Ly6C (Clone 1G7.G10, Miltenyi Biotec, Bologna, Italy), CD51 (Clone RMV-7, BD Biosciences), TLR2 (Clone 6C2, eBioscience) and TLR4 (Clone MTS510, Biolegend).

### 2.8. Macrophage Polarization and Treatment with EVs 

The yMΦ and aMΦ were polarized toward a pro-inflammatory M1 state adding 100 ng/mL Lipopolysaccharides (LPS) (Sigma) and 50 ng/mL of Recombinant Murine IFN-γ (Peprotech) to the complete medium for 24 h. After stimulation, the medium was discarded, replaced with complete medium, and the EVs derived from Hyp^INF^-hMSCs were added or not to 2ml cell cultures (0.25 µg/mL). The expression of the F4/80 Monoclonal Antibody, PE-Cy7 (Clone BM8, eBioscience), anti-mouse iNOS APC (Clone CXNFT, eBioscience), and CD38 Monoclonal Antibody, PE (Clone 90, Thermo Fisher Scientific) was evaluated by flow cytometry. 

### 2.9. RNA Extraction and Quantitative Real Time-PCR

The RNA was extracted from both yMΦ and aMΦ with RNeasy plus micro kit (Qiagen, Hilden, Germany). The RNA was retrotranscribed with an iScript cDNA synthesis kit (Biorad, Hercules, CA, USA). The expressions of TNF-α and CD38 were determined by quantitative Real Time-PCR and β2-microglobulin was considered to be housekeeping gene. Primers used in Real Time PCR were: TNF-α (Forward primer: 5′-AAA ATT CGA GTG ACA AGC CTG TAG C-3′; Reverse primer: 5′-GTG GGT GAG GAG CAC GTA G-3′); CD38 (Forward primer: 5′-TTG CAA GGG TTC TTG GAA AC-3′; Reverse primer: 5′-CGC TGC CTC ATC TAC ACT CA-3′); β2-microglobulin (Forward primer: 5′-CTG CTA CGT AAC ACA GTT CCA CCC-3′; Reverse primer: 5′-CAT GAT GCT TGA TCA CAT GTC TCG-3′). 

### 2.10. Proteome Profiler Array 

The proteins secreted by mBMSCs from the aged and young mice were analysed using the Proteome Profiler™ Mouse XL Cytokine Array Kit (R&D Systems, Minneapolis, MN, USA). In each sample, 111 proteins were evaluated, according to the manufacturer’s instructions. Next, 180 μg of proteins of CM from the young and aged mBMSCs (yCM and aCM, respectively) were incubated with an antibody cocktail and then left up to the membranes overnight at 4 °C. Different washing steps were repeated, streptavidin-HRP was added for 30 min and membrane chemiluminescence was developed in the dark room. ImageJ with the plugin of Protein Array Analysis was used to determine the intensity of each pixel and yCM-associated proteins were compared with aCM-associated proteins. A heat map was generated using GraphPad Prism (Graph Pad Software, Inc., San Diego, CA, USA). 

### 2.11. Evaluation of Oxygen Consumption Rate

The oxygen consumption rate (OCR) was measured inside a closed chamber at 25 °C with an amperometric electrode (Unisense Microrespiration, Unisense A/S, Aarhus, Denmark). For each experiment, 2 × 10^5^ cells or 2 μg EVs were resuspended in phosphate buffer saline (PBS) and permeabilized for 1 min with 0.03 mg/mL digitonin. Ten mM pyruvate plus 5 mM malate or 20 mM succinate were employed to stimulate the pathways composed of complexes I, III, and IV, or II, III, and IV, respectively [28].

### 2.12. F_o_F_1_ ATP-Synthase Activity Assay

The F_0_F_1_ ATP-synthase (ATP Synthase) activity was evaluated incubating 2 × 10^5^ cells or 2 μg EVs for 10 min at 25 °C in a medium containing: 50 mM Tris-HCl (pH 7.4), 50 mM KCl, 1 mM EGTA, 2 mM MgCl_2_, 0.6 mM ouabain (a Na/K ATPase inhibitor), 0.25 mM di(adenosine)-5-Penta-phosphate (an adenylate kinase inhibitor), and 25 μg/mL ampicillin (0.1 mL final volume). As for the OCR evaluation, 10 mM pyruvate plus 5 mM malate or 20 mM succinate were employed to stimulate complexes I, III, and IV or complexes II, III, and IV pathways. ATP synthesis was induced with the addition of 0.1 mM ADP. The reaction was monitored every 30 s for 2 min with a luminometer (GloMax^®^ 20/20 Luminometer, Promega, Milano, Italy), using the luciferin/luciferase ATP bioluminescence assay kit CLS II 8 Roche, Basel, Switzerland). For calibration, ATP standard solutions in a concentration range between 10^−8^ and 10^−5^ M were used [28].

### 2.13. P/O Ratio

The P/O value, an OxPhos efficiency marker, is calculated as the ratio between aerobic synthesized ATP synthesis and consumed oxygen. Efficient mitochondria display a P/O value of around 2.5 or 1.5 when the pathways led by complexes I or II are stimulated, suggesting a complete coupling between energy synthesis and respiration. Conversely, a lower P/O ratio indicates an uncoupled status in the OxPhos metabolism, which may contribute to the reactive oxygen species (ROS) production [29].

### 2.14. Cell Homogenate Preparation

The cells were centrifuged at 1000 rpm for 5 min, and the growth medium was removed. The pellet was washed in PBS twice and centrifuged again. The pellet was resuspended in Milli-Q water plus protease inhibitor and sonicated in ice twice for 10-s, with a 30-s break to prevent the warming, using the Microson XL Model DU-2000 (Misonix Inc., New York, NY, USA). The total protein content was estimated by the Bradford method [30].

### 2.15. ATP and AMP Intracellular Content Evaluation

The ATP was assayed spectrophotometrically following NADP reduction at 340 nm. The assay mix contained: 100 mM Tris-HCl (pH 8.0), 0.2 mM NADP, 5 mM MgCl_2_, and 50 mM glucose. Samples were analyzed before and after the addition of 3 μg of purified hexokinase plus glucose-6-phosphate dehydrogenase. AMP was assayed spectrophotometrically following NADH oxidation at 340 nm. The reaction medium contained: 100 mM Tris-HCl (pH 8.0), 5 mM MgCl_2_, 0.2 mM ATP, 10 mM phosphoenolpyruvate, 0.15 mM NADH, 10 IU adenylate kinase, 25 IU pyruvate kinase, and 15 IU of lactate dehydrogenase. ATP/AMP ratio was calculated as the ratio between the intracellular concentration of ATP and AMP, expressed in mM/mg of total protein [31].

### 2.16. Lactate Dehydrogenase Activity Assay

The lactate dehydrogenase (LDH) activity was assayed spectrophotometrically following NADH oxidation at 340 nm in the presence of 1 mM pyruvate and 0.2 mM NADH, buffered by 100 mM Tris-HCl, pH 7.4 [32].

### 2.17. Malondialdehyde Evaluation

Malondialdehyde (MDA), a lipid peroxidation marker, was evaluated by the thiobarbituric acid reactive substances (TBARS) assay. This test is based on the thiobarbituric acid (TBA) reaction with MDA, a breakdown product of lipid peroxides. The TBARS solution contained 26 mM thiobarbituric acid and 15% trichloroacetic acid (TCA) in 0.25 N HCl. To evaluate the MDA concentration, 50 μg of total proteins were dissolved in 300 μL of Milli-Q water and added with 600 μL of TBARS solution, and incubated at 95 °C for 60 min. After a 14,000-rpm centrifugation for 2 min, the supernatant was analyzed spectrophotometrically at 532 nm [33].

### 2.18. Statistical Analysis

The data were analyzed appropriately using *t*-test, one-way or two-way ANOVA, using Prism 9 Software. The data are expressed as mean ± standard deviation (SD) and are representative of at least three independent experiments. An error with a probability of *p* < 0.05 was considered significant.

## 3. Results

### 3.1. The Bone Marrow (BM) of Aged Mice Contains a Higher Number of Pro-Inflammatory Macrophages Compared to the Young Counterpart

Macrophages are central to the development of inflammaging [34]. We observed that the number of BM nucleated cells was significantly increased in the aged (78–104 weeks old) compared to the young (8–12 weeks old) mice (aBM and yBM, respectively) (*p* < 0.001) (Figure 1a). Inducible nitric oxide synthase (iNOS) is a key enzyme for the macrophage inflammatory response and is considered an important marker of pro-inflammatory macrophage activation [35]. The BM of the aged mice presented a significantly higher number of iNOS positive (iNOS^POS^) cells than the young BM (*p* < 0.001) (Figure 1b). Accordingly, macrophages, (MΦ) selected by Macrophage-Colony Stimulating Factor (M-CSF) from either aBM or yBM (aMΦ and yMΦ, respectively), presented significant differences. We first examined the expression of cell surface markers in yMΦ and aMΦ by flow cytometry (Figure 1c). We found that the percentage of F4/80, CD11c, and CD51 was similar in both cell populations, independent of age (Figure 1c). As already reported [36], the expression of lymphocyte antigen 6 complex, locus c (Ly6C) was three-fold higher in aMΦ than in the young counterpart (Figure 1c). Under basal conditions, the expression of TLR2 and TLR4 also showed differences between yMΦ and aMΦ, with TLR2 being 1.7-fold more expressed by aMΦ and TLR4 1.9-fold more expressed by aMΦ (Figure 1c). The mRNA level of the pro-inflammatory mediator TNF-α was measured and compared in aMΦ and yMΦ, in basal conditions and upon a 24 h stimulation with LPS and IFNγ, alone or in combination (Figure 1d,e). yMΦ responded to stimuli up-regulating the expression of TNF-α, even if only the synergistic effect exerted by LPS and IFN-γ induced a significant increase in comparison to control condition (*p* < 0.05) (Figure 1d). aMΦ are unresponsive to LPS stimulation, showing a level of TNF-α expression comparable to the control condition (Figure 1e) and confirming literature data reporting that ageing induces a metabolic dysregulation on BM-macrophages in response to LPS stimulation [23]. Ageing did not affect the TNF-α levels by aMΦ when the cells were stimulated with IFN-γ alone, whereas aMΦ responded to IFN-γ and LPS, significantly increasing the expression of TNF-α not only in comparison with CTRL (*p* < 0.05), but also with LPS (*p* < 0.01) (Figure 1e).

### 3.2. The Aged Microenvironment Induces an Accumulation of CD38^POS^ BM-Macrophages

During ageing, macrophages undergo a process of dysregulation in response to senescence stimuli, leading to a chronic inflammatory state [37]. One of the most used senescence-associated markers is senescence-associated (SA) lysosomal β-galactosidase activity (SA-β-gal), which is robustly enhanced in senescent cells as a result of increased lysosomal content [38]. However, one of the drawbacks in evaluating this marker is that even under normal physiological conditions, SA-β-gal activity is enriched in particular cell types, including mature macrophages [39]. Indeed, both yMΦ and aMΦ were positively stained by SA-β-gal, confirming that it is not a useful marker to detect senescent macrophages (Appendix A).

It has been recently reported that during ageing and in age-related diseases, a significant reduction in tissue nicotinamide adenine dinucleotide (NAD) occurs and, at tissue level, accumulated pro-inflammatory macrophages express a NAD consuming enzyme (CD38) [17]. To verify if the aged microenvironment could influence the overexpression of CD38 by BM-macrophages, we isolated the secretome from the bone marrow stromal cells (mBMSCs) of both the aged and young mice. After confirming the actual identity of the aged and young mBMSCs (Appendix A), we analyzed their conditioned media (aCM and yCM, respectively) by cytokine array, which allowed the contemporary analysis of 105 cytokines. We focused on proteins upregulated in aCM (Figure 2a) compared to yCM. The percentage of the inflammatory soluble factors CCL5, CCL12, CXCL1, CXCL2, LIX (CXCL5), CXCL16, osteoprotegerin and pentraxin-3 was higher in aCM than in the young counterpart, indicating that the aged BM environment is enriched of senescence associated secretory phenotype (SASP) factors. We considered whether the SASP from senescent BM stromal cells could promote CD38 expression in the macrophages. When the macrophages retrieved from the young mice were cultured for 24 h in the presence of aCM, they changed morphology, adopting a less elongated, ameboid shape, compared to the cells stimulated with yCM or α-MEM (standard condition) (Figure 2b). More importantly, we observed that the percentage of CD38+ yMΦ increased significantly when stimulated with aCM, compared to the cells maintained in a standard condition or stimulated with yCM (*p* < 0.05) (Figure 2c). These data provide in vitro evidence that the inflammatory cytokines comprising the SASP of senescent cells are key drivers of CD38 expression by macrophages.

### 3.3. Characterization of Hyp^INF^-hMSC-Derived EVs 

We have previously reported that the appropriate preconditioning of hMSCs, through an in vitro culture protocol that mimics an injury environment, improves their therapeutic efficacy [22]. In the present study, EVs have been isolated from the conditioned medium (CM) of hMSCs, maintained for 24 h in hypoxic (1% O_2_) and inflammatory (TNF-α and IL-1α) conditions (Hyp^INF^), as previously described [22]. The CM underwent low speed centrifugations (300× *g* and 2000× *g*) to remove possible cells/cell debris and apoptotic bodies, respectively. The resulting supernatants were then subjected to ultracentrifugation, first at 10,000× *g* (10K) to isolate the medium-sized EVs (mEVs), and then at 100,000× *g* (100K) for the smallest EVs (sEVs). The mEVs and sEVs were pooled in order to analyze the whole EV population released by Hyp^INF^-MSCs. Nanoparticle tracking analysis (NTA) indicated that the EVs presented a mean size of 153.6 ± 77.37 nm with a peak at 117.36 nm. Figure 3a shows a representative NTA analysis of EVs separated from three primary cultures of Hyp^INF^-hMSCs. We also estimated the concentration of vesicles contained in 1 μg of the EVs by BCA. Figure 3b indicates that the concentration contained in 1 μg of EVs was variable, ranging between 2.08 × 10^8^ and 4.98 × 10^8^. Western blot analysis was performed on the EVs and corresponding cell lysates, evaluating the expression of the two vesicle markers Syntenin-1 and Flotilin-1, and the negative control Grp94, a HSP90-like protein specialized for protein folding and quality control in the endoplasmic reticulum. As expected, Grp94 was expressed only by cell lysates (Figure 3c), while Syntenin-1, a specific EV marker, was expressed only by EVs (Figure 3c). Flotilin-1, a protein associated with membrane microdomains enriched in cholesterol and sphingolipids that plays a role in EV release, was expressed by both cell lysates and EVs (Figure 3c). EVs separated from Hyp^INF^-CM were also analyzed by flow cytometry to evaluate the expression of the tetraspanin family members CD81, CD63 and CD9, using a non-conventional flow cytometry approach based on the use of fluorescent dimensional beads of known diameters and of the fluorescent lipophilic tracer CFDA-SE, that is able to passively diffuse within vesicles and interact with intra-vesicular enzymes at room temperature (RT) [27]. The CFDA-SE positive EVs expressed CD81, CD63 and CD9 at different levels. The mean fluorescence intensity (MFI) associated with the considered markers was calculated in 3 independent experimental replicates. As shown in the histograms presented in Figure 3d,e, CD81 was the most expressed antigen on the vesicle surface with a mean of 12.407 ± 5.328, while CD63 (6.577 ± 0.366) and CD9 (2.198 ± 0.314) were less expressed.

### 3.4. EVs Released by Hyp^INF^-hMSCs Exert an Anti-Inflammatory Role on Aged Macrophages

In order to evaluate whether the EVs released by Hyp^INF^-hMSCs were able to modulate the inflammatory state of macrophages, we tested their effect in vitro on both young and aged cells, not stimulated (yCtrl and aCtrl, respectively) or upon stimulation for 24 h with LPS and IFN-γ (M1 condition) (yM1 and aM1, respectively). After 24 h, the culture medium of the control and M1-stimulated cells was replaced, for an additional 24 h, with 2 mL of serum-free IMDM medium in presence of EVs (0.25 μg/mL). While at mRNA level there were no significant differences in the expression of TNF-α between yCtrl and aCtrl (Figure 4a), this difference was statistically significant at the protein level (*p* < 0.05) (Figure 4b). Stimulation toward an M1 phenotype was effective for aged macrophages, leading to a significant increase in TNF-α expression at both the mRNA (*p* < 0.0001) (Figure 4a) and protein level (*p* < 0.001) (Figure 4b), compared to control conditions. In addition, in this case, the aM1 macrophages expressed TNF-α in a statistically greater manner than the young counterpart, at both the mRNA (*p* < 0.001) (Figure 4a) and protein level (*p* < 0.0001) (Figure 4b), indicating a more inflamed phenotype not only in basal conditions, but also in response to inflammatory stimuli. Interestingly, when the aged M1 macrophages were treated with EVs, the protein levels of TNF-α were significantly down-regulated (aM1 vs. aM1+EVs, *p* < 0.001) (Figure 4b). The EV-mediated anti-inflammatory potential was confirmed by flow cytometry, evaluating the expression of two specific markers by young and aged M1 macrophages, treated or not with EVs. The selected markers were F4/80, a pan-macrophage marker, and iNOS, which is typically expressed by mouse M1 macrophages. Figure 4c shows that both the young and aged macrophages were F4/80-positive (F4/80^POS^) and expressed high levels of the intracytoplasmic marker iNOS, without significant differences among the two considered cell populations (Figure 4c, upper panels). Within the F4/80^POS^ iNOS^POS^ cells, we observed two subpopulations, NOS^BRIGHT^ and iNOS^DIM^, indicating the existence of two iNOS2^POS^ subpopulations characterized by a different antigenic density (Figure 4c). When the M1 macrophages were treated for 24 h with EVs, following the same experimental plan described at the beginning of the paragraph, the percentage of F4/80^POS^ iNOS^BRIGHT^ cells in the aged population was significantly decreased (*p* < 0.05) (Figure 4d). 

We also investigated whether ageing could lead to a dysregulation of the NADase CD38 expression by bone marrow-macrophages and whether Hyp^INF^ MSC-derived EVs were able to dampen this phenomenon. As shown in Figure 4e (upper panels) and Figure 4f, in basal conditions, the percentage of old F4/80^POS^ CD38^POS^ aged macrophages was significantly higher than the young counterpart (*p* < 0.001). More importantly, the treatment for 24 h with 0.25 μg/mL EVs induced a significant decrease in CD38-expressing aged macrophages in comparison to the corresponding control condition (*p* < 0.05) (Figure 4e, bottom panels and Figure 4f). This confirmed the hypothesis that aged macrophages are characterized by a basal level of this NAD-consuming enzyme and EVs released by pre-conditioned hMSCs exert a specific anti-senescent effect.

### 3.5. Hyp^INF^ hMSCs-Derived EVs Counteract Age-Associated Mitochondrial Dysfunction on Macrophages

Ageing decreases the mitochondrial transmembrane potential, as indicated by the analysis of the oxidative phosphorylation (OxPhos) activity induced by pyruvate plus malate. Under basal condition (CTRL), aged MΦ showed a significant decrement of the oxygen consumption rate (OCR) (*p* < 0.0001) accompanied by a significant reduction in ATP synthesis (*p* < 0.0001), in comparison to young MΦ (Figure 5a,b). The stimulation toward a M1 phenotype induced uncoupling between oxygen consumption and ATP synthesis, independent of age (Figure 5a,b). Indeed, the M1 stimulation was characterized by an inefficient oxidative phosphorylation, as also indicated by a reduction in the P/O ratio value, in comparison to the corresponding CTRL conditions (*p* < 0.0001) (Figure 5c). Under basal conditions, the treatment with EVs partially rescued the age-associated dysfunction of OCR (*p* < 0.01) and ATP synthesis (*p* < 0.0001) (Figure 5a,b), and this was accompanied by an improvement, compared to CTRL values, of the coupling between oxygen consumption and ATP synthesis observed in M1 conditions, as shown by the P/O ratio (Figure 5a,b). In detail, in aged MΦ, the P/O ratio values were significantly increased in the control condition (*p* < 0.0001) or completely rescued to CTRL values in the M1 condition (*p* < 0.0001) upon EV treatment (Figure 5c). In the young counterpart, the addition of EVs did not prompt significant changes in the P/O ratio values in the control condition, whereas it induced a significant increment in the P/O ratio values in the M1 condition (*p* < 0.0001) (Figure 5c). A similar trend was observed when OxPhos activity was induced by succinate (Figure 5d–f). As the loss of mitochondria efficiency can cause an enhancement of the anaerobic metabolism, the activity of the lactate dehydrogenase (LDH) has been evaluated as a marker of the anaerobic glycolytic pathway. The data show an increase in LDH activity in aged MΦ, already in basal conditions, compared to young MΦ (MDA: *p* < 0.0001; LDH: *p* < 0.0001) (Figure 5g), and an increment upon M1 stimulation, independent of age (MDA: *p* < 0.0001 for both young and aged MΦ; LDH: *p* < 0.0001 for both young and aged MΦ) (Figure 5g). EV treatment dampened this phenomenon, decreasing the LDH values in aged MΦ in the control (*p* < 0.0001) and M1 (*p* < 0.0001) conditions, while in the young cells, the decrease was significant only upon M1 stimulation (*p* < 0.0001) (Figure 5g). Moreover, as the uncoupled OxPhos metabolism is often associated with an increment in the oxidative stress production [40], the level of malondialdehyde (MDA), a marker of lipid peroxidation, has been evaluated (Figure 5h). Similar to LDH, the MDA concentration was significantly increased in aged MΦ, already in basal conditions, compared to young MΦ (*p* < 0.0001) (Figure 5h), and increased upon M1 stimulation, independent of age (MDA: *p* < 0.0001 for both young and aged MΦ) (Figure 5h). 

To determine the energy status of young and aged MΦ, the intracellular concentration of ATP and AMP was measured, and the ratio ATP/AMP was calculated (Figure 6a–c). The energy status changed with age, showing a significant decrease in basal conditions (*p* < 0.0001) (Figure 6c). This difference was even more evident when both young and aged cells were polarized toward a M1 state, comparison to control conditions (*p* < 0.0001 for both young and aged MΦ) (Figure 6c). The cause of the decrement of ATP/AMP was partially due to the significant increment of the AMP concentration in the aged cells and upon M1 stimulation (Figure 6a), accompanied by an increment of ATP in the same conditions (Figure 6b). Treatment with EVs significantly increased the energy status of both young and aged MΦ, both in control and M1 conditions (young: CTRL vs. CTRL+EVs: *p* < 0.05; young: M1 vs. M1+EVs: *p* < 0.0001; aged: CTRL vs. CTRL+EVs: *p* < 0.0001; aged: M1 vs. M1+EVs: *p* < 0.0001) (Figure 6c).

To determine whether the positive effect induced by the EV treatment on macrophages could be caused by the energetic characteristics of the EVs themselves, the oxygen consumption rate, ATP synthesis and P/O ratio of the EVs were measured (Figure 6d–f). The data show that in the presence of either pyruvate plus malate (P/M) or succinate, EVs can conduct an aerobic metabolism, consume oxygen (Figure 6d), and synthesize ATP (Figure 6e). This energy metabolism also turned out to be coupled, as the P/O value (Figure 6f) is virtually identical to the literature values for coupled mitochondria, for both the P/M and succinate [41].

## 4. Discussion

The consequences of aging are the systemic activation of inflammatory and innate immune cells, namely inflammaging, in response to chronic stress signals in both physiological and pathological conditions [42]. Age-associated inflammation is characterized by multivariable low-grade, chronic and systemic responses that exacerbate the aging process and the age-related diseases [43]. The search for therapeutic approaches and innovative strategies to counteract or, at least, keep inflammaging under control, is still at an early stage. This is probably due to a partial understanding of the mechanisms, evaluation methods, research models and possible interventions underlying inflammaging [4]. 

The secretome, and in particular the fraction composed by EVs, derived from MSCs can stimulate tissue repair, modulating the inflammatory and immune responses [44,45,46]. This is particularly true when MSCs are appropriately preconditioned to survive in hostile conditions, enhancing their regulatory activities [22]. Indeed, we previously demonstrated that the MSC secretome was strongly modulated by the simultaneous stimulation with hypoxia and pro-inflammatory cytokines (TNF-α and IL-1α) (Hyp^INF^ condition), used to mimic the harsh environment present at the site of injury. The paracrine activity of preconditioned cells was enriched in factors involved in the interaction with innate immune cells and in tissue remodelling and repair [22]. Here, we evaluate whether EVs released by Hyp^INF^ MSCs are able to exert an anti-inflammatory role in the context of inflammaging, using bone marrow-macrophages as prototype cells, as their age-associated dysregulation and chronic activation is a primary contributor to inflammaging [16]. We observed that the bone marrow of aged mice contained a higher number of nucleated and iNOS-positive cells than the young counterpart, and the pro-inflammatory responses were higher in macrophages obtained from aged mice. iNOS is a hallmark molecule of pro-inflammatory M1 macrophages [35]; therefore, we specifically focused on bone marrow-macrophage characterization. We found elevated pro-inflammatory factor levels expressed by aged macrophages. High levels of TNF-α, iNOS and CD38 have been observed, not only when aged macrophages were challenged with IFN-γ and LPS, as also described in literature [47], but also in basal conditions. These results suggest that during aging, macrophages are in a pre-activated state, already in resting conditions, and this enhances their responsiveness to pro-inflammatory stimuli. Interestingly, aged macrophages were hyporesponsive to LPS stimulation, confirming previously published results [23] and suggesting that metabolic dysregulations play a key role in aged macrophage function. The increased expression of CD38 by the aged macrophages raised our interest, as this marker is necessary not only for immune cell activation [48], but also for the age-related NAD decline at tissue level. Indeed, CD38 is a multifunctional enzyme that uses NAD as a substrate to generate second messengers and is therefore implicated in the consumption of tissue NAD during the aging process [18]. It has been reported that in epididymal white adipose, liver and bone marrow tissues, pro-inflammatory M1 macrophages exhibit increased CD38 expression and enhanced NADase activity [17,49]. Interestingly, we observed an increased expression of CD38 by bone marrow macrophages derived from the aged (72–108 weeks old) mice also in basal condition, compared to young ones, and this reinforced the idea that aging leads to macrophage pre-activation. Moreover, we confirmed previously published data reporting that, during aging, CD38 expression by tissue resident macrophages is driven by the senescence-associated secretory phenotype (SASP) of senescent cells [12]. In our experimental set-up, the aged microenvironment represented by the bone marrow stromal compartment was responsible for the accumulation of CD38-positive BM-macrophages. Specifically, the conditioned medium, enriched of SASP factors, released by the bone marrow stromal cells of aged mice was able to induce the expression of CD38 by young macrophages, corroborating the idea that a senescence bystander effect could be directly linked to macrophage proliferation and activation. This could also partially justify the observed age-related increase in the number of bone marrow-nucleated cells and macrophages. Given these results, we wondered whether Hyp^INF^ MSC-derived EVs were able to dampen age-associated inflammation. Firstly, we noticed that the in vitro treatment with EVs at a concentration of 0.25 μg/mL was capable of lowering the expression levels of the above-mentioned inflammatory markers by aged macrophages, both at mRNA and protein levels and to modulate the expression of the senescent marker CD38. 

Regarding energy metabolism, we observed an age-related impairment of mitochondrial function in the aged macrophages, where we found an induced metabolic switch towards anaerobic glycolysis. Specifically, under basal conditions, aging led to a significant lowering of the oxygen consumption rate, accompanied by a reduction in ATP synthesis by BM-macrophages. The stimulation toward a M1 phenotype induced uncoupling between oxygen consumption and ATP synthesis, independent of age. As impaired mitochondria cause cumulative oxidative stress that is strictly linked to the aging process and inflammation [50,51], the search for strategies for reducing or preventing these phenomena is desirable. Therefore, we have evaluated the EVs treatment effects on mitochondrial metabolism, observing a restoration of the OxPhos function and efficiency and the anaerobic glycolytic rate reduction. In addition, our preliminary data suggest that these effects are caused by the energetic characteristics of the EVs themselves. Indeed, Hyp^INF^-EVs were able to conduct an aerobic metabolism, consume oxygen, and synthesize ATP, as already observed for EVs derived from umbilical cord mesenchymal stem cells [52].

Although preliminary, and requiring confirmation with appropriate in vivo experimental models, our results could pave the way to the development of therapeutic strategies based on the use of MSC-EVs to confer beneficial effects in the landscape of aging cells and organs. Targeting inflammaging via MSC-EVs could be considered an emerging potential solution to overcome the challenges in age-related disorders (i.e., osteoarthritis).

## Figures and Tables

**Figure 1 cells-11-03695-f001:**
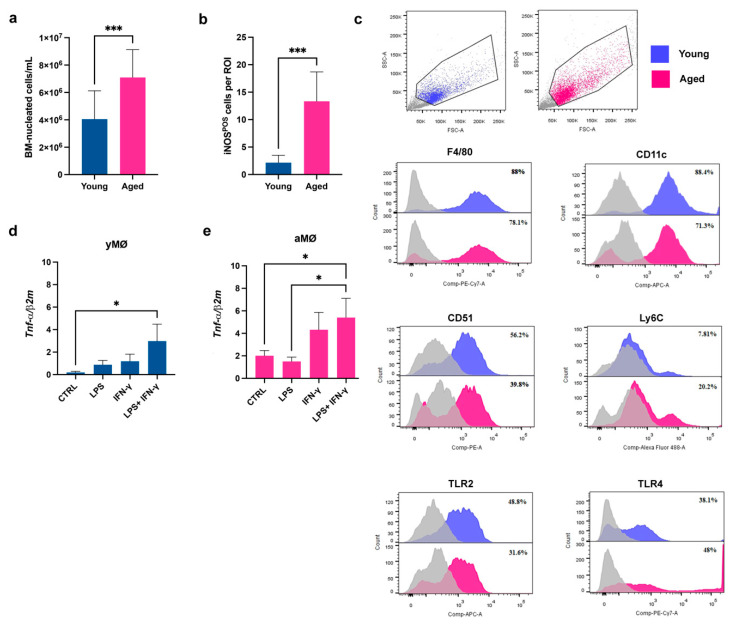
**Effects of aging on bone marrow (BM)-nucleated cells and BM-macrophages.** (**a**) Quantification of nucleated cells from the BM of young (*n* = 14) and aged (*n* = 13) mice. Data are presented as mean ± SD. *** *p* < 0.001 (Unpaired *t*-test with Welch’s correction). (**b**) Quantification of iNOS-positive (iNOS^POS^) cells stained by immunohistochemistry and detected in the BM of young and aged mice. iNOS^POS^ cells have been counted considering 5 different regions of interest (ROI)/BM. Data are presented as mean ± SD. *** *p* < 0.001 (Unpaired *t*-test). (**c**) Representative flow cytometry analysis of either young and aged BM-macrophages (yMΦ and aMΦ, respectively) maintained in complete medium for 7 days. Dot plots indicate the dimensional gates selected for both young and aged macrophages. In the histograms, areas under blue and red lines identify yMΦ and aMΦ reacting with F4/80, CD11c, Ly6C, CD51, TLR2 and TLR4. Areas under the grey lines indicate the interactions of cells with corresponding non-reactive immunoglobulin of the same isotype. (**d**,**e**) Quantitative Real Time PCR for the expression of *TNF-α* levels in yMΦ (**d**) and aMΦ (**e**) cultured for 24 h in standard medium (CTRL) or under stimulation with LPS, IFNγ, alone or in combination (LPS+IFN-γ). Histograms represent the mean ± SD of 3 independent experimental replicates. * *p* < 0.05 (One-way Anova).

**Figure 2 cells-11-03695-f002:**
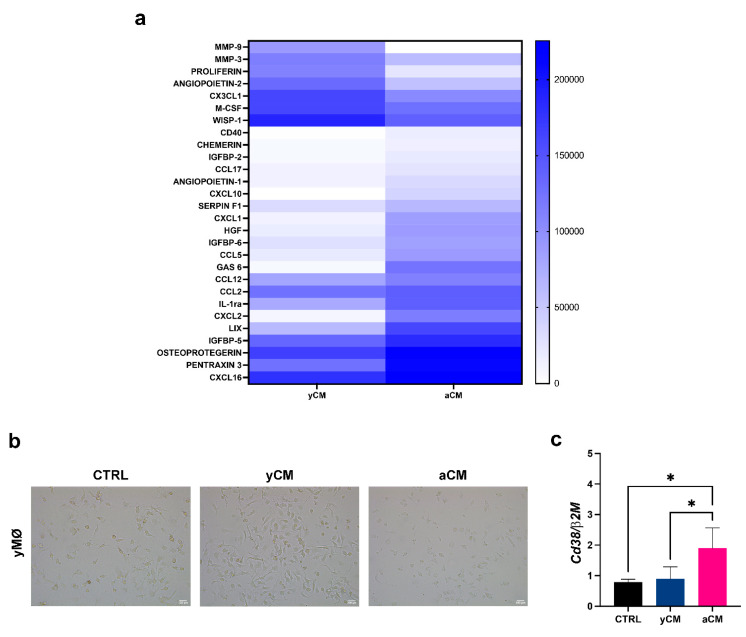
**Effects of an aged BM microenvironment on young macrophages.** (**a**) Heatmap of the expressed proteins revealed by the proteome profiler analysis of the conditioned medium (CM) derived from young and aged mouse bone marrow stromal cells (yCM and aCM, respectively). (**b**) Representative image of young macrophages cultured for 24 h in serum-free α-MEM medium (CTRL), or in presence of either 50 μg of yCM or 50 μg of aCM. (**c**) Quantitative Real Time PCR for the expression of *Cd38* levels in yMΦ maintained for 24 h in standard medium (CTRL) or in presence of either 50 μg of yCM or 50 μg of aCM. * *p* < 0.05 (One-way Anova).

**Figure 3 cells-11-03695-f003:**
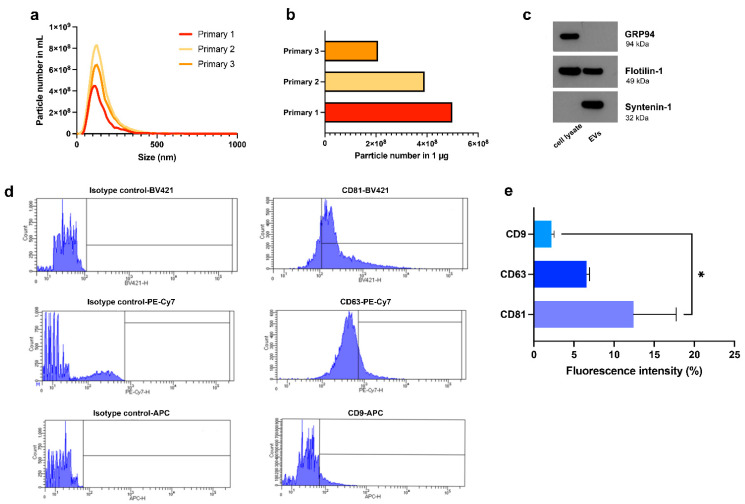
**Characterization of Hyp^INF^ MSC-derived EVs.** (**a**) Representative NTA analysis of EVs derived from 3 independent primary cultures of Hyp^INF^ MSCs, representing the particle number/μL and the EV size. (**b**) The histogram indicates the particle number present in 1 μg of EVs (calculated by BCA assay) from 3 independent primary cultures of Hyp^INF^ MSCs. (**c**) Western blot analysis performed on Hyp^INF^ MSC-derived EVs and corresponding cell lysate. (**d**) Representative flow cytometry analysis showing the expression of CD81, CD63 and CD9 (right panels) and corresponding isotype controls (left panels) by Hyp^INF^ MSC-derived EVs. (**e**) Quantification of CD9-, CD63-, and CD81-positive events falling within the CFDA-SE gate by EVs. Data are presented as ratio between mean fluorescence intensity (MFI) of cells stained with a specific antibody and MFI of correspondent isotype control (relative MFI). Data are representative of at least three independent experiments. * *p* < 0.05 (One-way Anova).

**Figure 4 cells-11-03695-f004:**
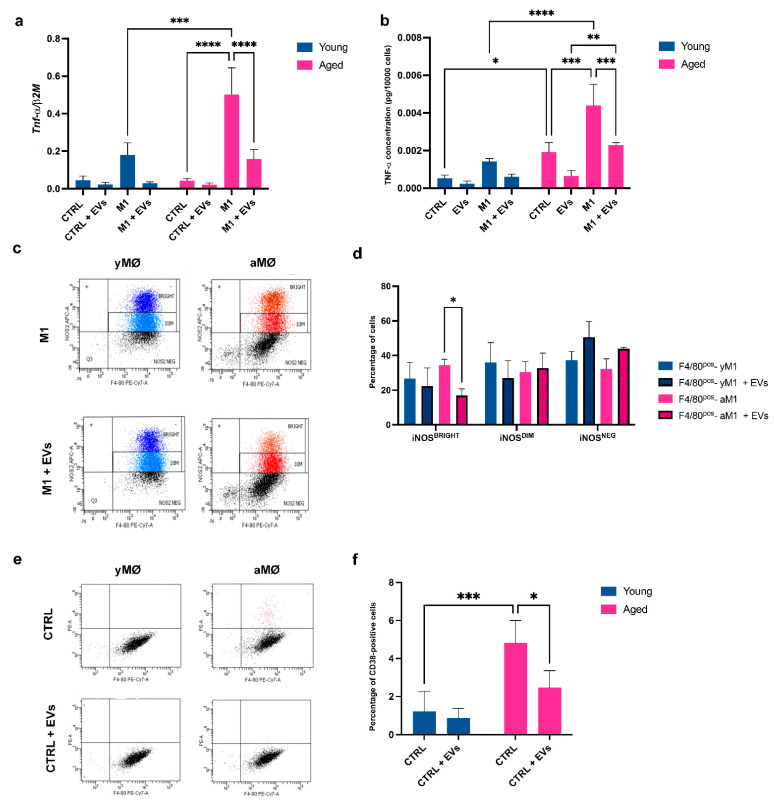
**Anti-inflammatory activity of Hyp^INF^ MSC-derived EVs.** (**a**) Quantitative Real Time PCR analysis evaluating the expression levels of *Tnf-α* in yMΦ and aMΦ maintained in 2 mL standard medium (CTRL) or in presence of the stimuli LPS and IFN-g (M1), with or without the addition of Hyp^INF^ MSC-derived EVs (0.25 μg/mL) for 24 h (CTRL+EVs and M1+EVs, respectively). Bars indicate mean ± SD from six independent experiments. *** *p* < 0.001, **** *p* < 0.0001 (Two-way ANOVA). (**b**) Concentrations of TNF-α measured by ELISA in culture supernatant from CTRL, CTRL+EVs, M1 and M1+EVs yMΦ and aMΦ. Bars indicate mean ± SD from six independent experiments. * *p* < 0.05, ** *p* < 0.01, *** *p* < 0.001, **** *p* < 0.0001 (Two-way ANOVA). (**c**) Representative flow cytometry analysis of yMΦ and aMΦ under M1 stimulation and treated or not with 0.5 μg EVs for 24 h. MΦ were labelled with specific anti-F4/80 and anti-iNOS antibodies. (**d**) The histogram shows the percentage of iNOS^Bright^, iNOS^Dim^ and iNOS^Neg^ yMΦ and aMΦ that were F4/80-positive upon stimulation toward M1 and treated with or without EVs. Bars indicate mean ± SD from three independent experiments. * *p* < 0.05 (Ordinary two-way ANOVA). (**e**) Representative flow cytometry analysis of yMΦ and aMΦ maintained in CTRL condition and treated or not with 0.25 μg/mL EVs for 24 h. MΦ were labelled with specific anti-F4/80 and anti-CD38 antibodies. (**f**) The histogram shows the percentage CD38-positive yMΦ and aMΦ maintained in CTRL condition and treated or not with 0.25 μg/mL EVs for 24 h. Bars indicate mean ± SD from three independent experiments. * *p* < 0.05, *** *p* < 0.001 (Two-way ANOVA).

**Figure 5 cells-11-03695-f005:**
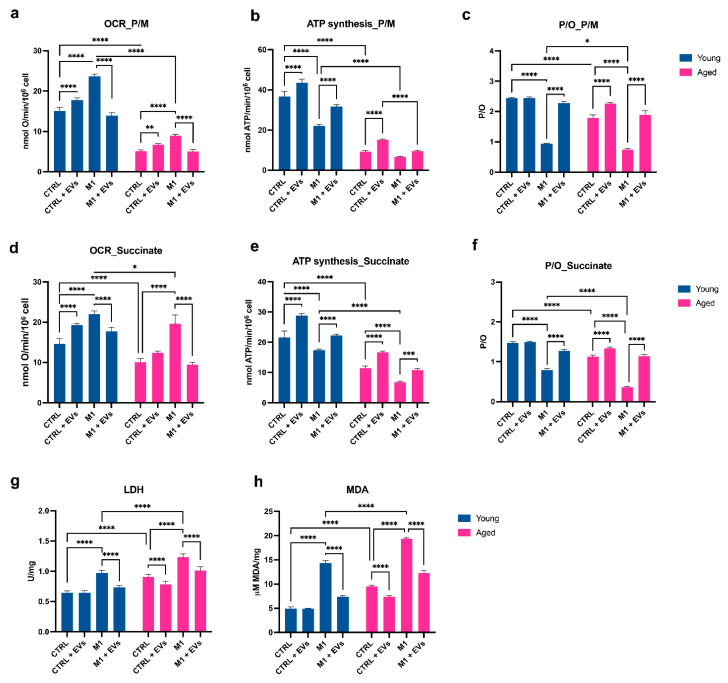
**Hyp^INF^ MSC-derived EVs sustain the mitochondrial metabolism of aMΦ.** All the biochemical measurements refer to yMΦ and aMΦ maintained in 2 mL of standard medium (CTRL) or in presence of the stimuli LPS and IFN-γ (M1), with or without the addition of Hyp^INF^ MSC-derived EVs (0.25 μg/mL) for 24 h (CTRL+EVs and M1+EVs, respectively). Oxygen consumption rate stimulated with pyruvate plus malate (**a**) or succinate (**d**); ATP production by F_0_–F_1_ ATP synthase stimulated with pyruvate plus malate (**b**) or succinate (**e**); macrophage P/O ratio (marker of OxPhos efficiency) after pyruvate plus malate (**c**) or succinate (**f**) stimulation; lactate dehydrogenase activity (LDH) activity (**g**); malondialdehyde (MDA) intracellular concentration (**h**). Bars indicate mean ± SD from three independent experiments. * *p* < 0.05, ** *p* < 0.01, *** *p* < 0.001, **** *p* < 0.0001 (Two-way ANOVA).

**Figure 6 cells-11-03695-f006:**
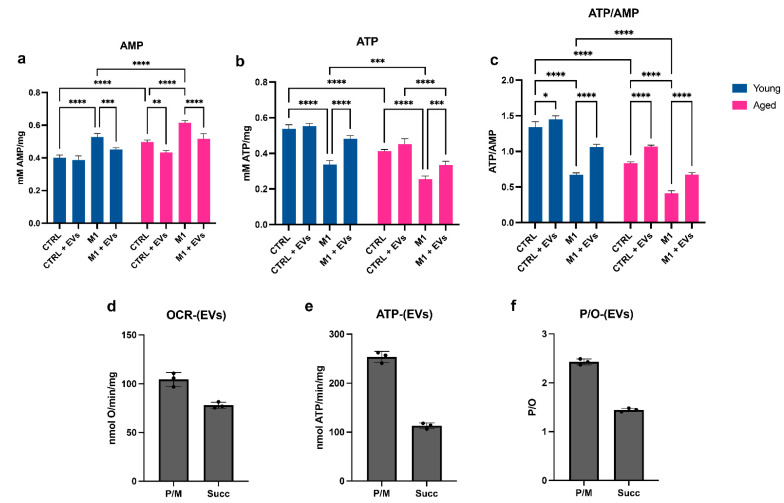
**Modulation of ATP and AMP intracellular concentrations and consequent energy status in yMΦ and aMΦ.** All the biochemical measurements refer to yMΦ and aMΦ maintained in 2 mL of standard medium (CTRL) or in presence of the stimuli LPS and IFN-γ (M1), with or without the addition of Hyp^INF^ MSC-derived EVs (0.25 μg/mL) for 24 h (CTRL+EVs and M1+EVs, respectively). (**a**) Intracellular ATP content. (**b**) Intracellular AMP content. (**c**) ATP/AMP ratio, as a marker of cellular energy status. Each graph represents three independent experiments and data are expressed as mean ± S.D. * *p* < 0.05, ** *p* < 0.01, *** *p* < 0.001, **** *p* < 0.0001 (Two-way ANOVA). (**d**–**f**) All measurements refer to Hyp^INF^-EVs. Oxygen consumption rate stimulated with pyruvate plus malate or succinate (**d**); ATP production by F_0_–F_1_ ATP synthase stimulated with pyruvate plus malate or succinate (**e**); P/O ratio (marker of OxPhos efficiency) after pyruvate plus malate or succinate (**f**) stimulation.

## Data Availability

Not applicable.

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
