# Peer review of "Preconditioned Mesenchymal Stromal Cell-Derived Extracellular Vesicles (EVs) Counteract Inflammaging"

_cells, 2022, doi:10.3390/cells11223695_

Round 1
Reviewer 1 Report
The authors describe the potential anti-inflammatory role of extracellular vesicles isolated from mesenchymal stromal cell-derived preconditioned with hypoxia inflammatory cytokines. The authors have already done earlier work showing the benefits of primed MSC secretome.
I believe the data is useful for readers and experts in inflammation who search for strategies to mitigate the inflammatory process.
The article is well organized and easy to understand with all the sections well-developed.
The methods are well explained.
The results are clearly explained.
I have some comments for the authors:
1) page 6 “M1 macrophages were treated or not with 0.5 μg/well of EVs” How did the authors find this value to be the optimum for rescue? Did the authors perform a previous dose-response test?
2) Page 9 “aMφ are responsive to LPS stimulation, showing a level of TNF-a expression comparable to the control condition (Figure 1e) and confirming literature data reporting that aging induces a metabolic dysregulation on BM-macrophages in response to LPS stimulation [23].”
aMφ do not respond to LPS treatment alone, indeed there is a slight reduction in TNF-a levels. The authors also refer to the metabolic dysregulation of BM-macrophages several times in the manuscript but don’t explain which “metabolic dysregulation” could be involved in this lack of response.
3) page 13 ”mEVs and sEVs were pooled in order to analyze the whole EV population released by HypINF-MSCs.”
The authors treated macrophages with pure small EVs or a mix of small and medium EVs? Also, the data presented for NTA are related to sEVs only?
4) page 13 “while Syntenin-1, a specific EV marker, was expressed only by EVs (Figure 3c).”
EVs are enriched in Syntenin-1, highlighting its use as sEVS biomarker but it should appear in cell lysates.
5) In figure 3 b the results are intriguing as the same amount of protein concentration results in a huge variation in the EVs number. How do the authors explain these differences?
6) Figure 5 – aMφ M1 state treated with EVs shows a decrease in OCR upon stimulation with pyruvate plus malate or succinate which does not correlate with ATP production by Fo-F1 ATP synthase. Can the authors explain these contradictory results?
Author Response
Reviewer 1:
- Thanks to the Reviewer for this comment. We selected the dose of 0.5 mg EVs/well, since we previously reported that this EV amount was able to exert a pro-angiogenic and pro-proliferative effect both in vitro and ex vivo (doi: 10.1002/sctm.21-0107). So, we decided to maintain the same dose also to investigate the EV-mediated anti-inflammatory effect.
- We are sorry for this misunderstanding. We realized we had written “aMφ are responsive to LPS stimulation, showing a level of TNF-a expression comparable to the control condition (Figure 1e)…” instead of writing “aMφ are unresponsive to LPS stimulation, showing a level of TNF-a expression comparable to the control condition (Figure 1e)…”. We have corrected the text (pg. 9 of the revised version of the manuscript). Regarding the unresponsiveness to LPS stimulation by aged macrophages, it has been reported to be associated with a lack of shift from oxidative phosphorylation to glycolysis, as reported in the cited reference [23]. Moreover, it has been published that the critical regulatory metabolites succinate, γ-aminobutyric acid, arginine, ornithine, and adenosine were not increased in LPS-stimulated aged macrophages, contrary to the young counterpart.
- We treated macrophages with a pool of small- and medium-EVs, as indicated in the “Materials and Methods” section, “2.6 Extracellular vesicle (EV) separation and characterization” paragraph. We have decided to pool the two subpopulations because their dimension is in part overlapping, and we did not want to exclude any vesicle population from the analysis. In any case, in the present manuscript and in previous manuscripts (doi: 10.1002/sctm.21-0107; doi: 10.1016/j.biomaterials.2020.120633), we observed that the pool was enriched in small-EVs.
- As previously reported by our group (doi: 10.1002/sctm.21-0107), syntenin-1 is not expressed by human adipose-MSCs, contrary to bone marrow-derived MSCs. Only EVs derived from adipose-MSCs express this marker.
- Thanks to the Reviewer for this comment. We reported these data to show that the same number of hMSCs derived from different primary cultures can release different EV amounts. So, we think that these variations are linked to the variability associated with each primary culture.
- We thank the Reviewer for the question. Indeed, the results presented in Figure 5 show that treatments with EVs on M1 macrophages reduce Oxygen Consumption Rate (OCR) and increase levels of ATP synthesis for both young and aged samples. However, this apparent discrepancy can be explained by the OxPhos efficiency increment, probably due to the EVs treatment, as suggested by P/O values enhancement. The P/O value represents a mitochondrial energy efficiency marker, as it is the ratio between the number of ATP molecules synthesized by the FoF1 ATP synthase and the oxygen atoms consumed by the electron transport chain. P/O ratios around 2.5 and 1.5 indicate the maximum OxPhos efficiency when pathways led by Complex I or Complex II have been activated, respectively (doi: 10.1016/j.bbabio.2004.09.004). The maximum mitochondria efficiency is reached when oxygen consumption is completely devoted to ATP synthesis. In this condition, the oxygen consumption rate follows the "rhythm" of FoF1 ATP synthase that regulates the speed of the whole process. Conversely, in uncoupled conditions, the respiration rate dramatically increases because it is no longer under the control of ATP synthase, and the P/O values are lower than the reference values. Therefore, referring to the data reported in Figure 5, we can observe that M1 macrophages under basal conditions exhibit elevated respiration compared to the ATP synthesis, indicating a high uncoupling, as shown by the low P/O values. On the contrary, the treatment with EVs on M1 macrophages slows down oxygen consumption and maintains a stable ATP production, restoring the coupled condition, as shown by the P/O ratio around 2.5 and 1.5.

Reviewer 2 Report
EVs from mesenchymal stromal cells are promising tools to treat patients with various diseases. Gorgun and coworkers present a few interesting findings on the role of EVs to meliorate aging effects of murine macrophages. Therefore, they generated EVs from hypoxia/IFN-g conditioned stromal cells and found that such EVs were able to reduce the mitochondrial dysfunction and the concentration of TNF-a, iNOS and the NADase CD38, which are typical high in aged macrophages. They suggest that a clinical application of EVs from hypoxia-stimulated MSCs might be beneficial in age-related diseases.
The manuscript is a little difficult to read, because the focus is not always clear and the results deserve an improved explanation. As it stands, I am not always convinced that the authors drew the right conclusions. Also, the description of the methods needs improvement (see also below).
Detailed comments:
Methods should be improved, examples:
- high concentrations of antibiotics as written can be cytotoxic, i.e. 1% penicillin + 1% streptomycin (10 mg/mL+ 10 mg/mL). Maybe a stock solution was used. Standard in cell culture is: 100 IU and 100 µg/mL, respectively.
- Same is true for glutamine, in general 2 mM in cell culture medium
- The authors often applied ‘0.5 µg EVs’ but did not indicate the volume. This is not helpful as it stands.
Results
First paragraph, incorrect subtitle: the percentage was not determined but the number as written in the text.
Figure 1 and results: To support the findings in the results part, the author should add gates to illustrate the relevant cell population in (C). It is difficult to see the claimed differences in %-age (mainly TLR2,4).
Figure 1 A,B have a high SD. The LPS effect in young mice is much stronger than in old mice despite less TLR4. Was the experiment in 1C with the TLR2/4 expression reproduceable? There are only weak difference and only one FACS is shown.
The conclusions drawn from Fig 1 D and E are difficult to understand. Particularly the last sentence is not supported by the Figure or misleading.
Figure 2 and Figure Legend: What is the concentration of yCM and aCM? Was CM depleted of EVs or were EVs + soluble factors determined? Do soluble factors as determined in Figure 2A, contribute to the effects described in Fig. 3-6? Can such important factors functionally bind to EVs (corona) and mediate the anti-aging effects? This possibility should be discussed.
Figure 4 and Results: What concentration of EVs was applied (0.5 µg / xx mL)? The authors should explain the experiments better. As it appears to me, the readout in aged Mφ is 2-3 x stronger than in yMφ, i. e. TNFa: CTRL, EVs, M1, M1+EVs and CD38: CTRL and CTRL+EVs. EVs from MSCs reduce the effect to 50% of the non-treated control, in both, aged and young Mφ. So, the efficacy of EVs might be very similar in both systems, aged versus young. The authors might compare the %-age of change with CTRL as 100%.
Does hypoxia matter and to what extent? Only one part of the experiment was performed under hypoxia (generation of EVs). Is there also an effect with normoxia-borne EVs? The author should discuss this issue.
Figure 5 and Results: An interesting finding that EVs alone are able to produce energy.
Author Response
Reviewer 2:
Detailed comments:
Materials and Methods:
- We agree with the Reviewer, and we corrected the “Materials and Methods” section, paragraphs 2.2, 2.3, 2.4 and 2.7.
Results:
- We agree with the Reviewer’s comment, and we modified the subtitle of the first paragraph, from “3. The bone marrow (BM) of aged mice contains a higher percentage of pro-inflammatory macrophages compared to the young counterpart” to “3. The bone marrow (BM) of aged mice contains a higher number of pro-inflammatory macrophages compared to the young counterpart”.
- We realized to have written an incorrect sentence regarding figure 1d and 1e, since aged macrophages are unresponsive to LPS stimulation. We are sorry for this mistake, and we have corrected it in the revised version of the manuscript (please see also Reviewer 1, point 2).
Figure 1 and results:
We added the dimensional gates of both young and aged macrophages in figure 1c, as requested.
Figure 1a, b:
We are aware that the standard deviations referred to graphs in figure 1a and 1b is high despite experimental replicates, and we think this is probably due to the variability associated with the use of primary cultures and not cell lines. The LPS effect is stronger in young than in aged macrophages, as also indicated in the corresponding description of the results of the revised version of the manuscript. The data regarding TLRs is reproducible, and we did not observe particularly significant differences among young and aged macrophages, as also reported in some papers (doi: 10.1016/j.mad.2005.07.009).
“The conclusions drawn from Fig 1 D and E are difficult to understand. Particularly the last sentence is not supported by the Figure or misleading”.
As also indicated for Reviewer 1, we are sorry for the misunderstanding regarding the results related to figure 1d and 1e. We realized we had written “aMφ are responsive to LPS stimulation, showing a level of TNF-a expression comparable to the control condition (Figure 1e)…” instead of writing “aMφ are unresponsive to LPS stimulation, showing a level of TNF-a expression comparable to the control condition (Figure 1e)…”. We have corrected the text (pg. 9 of the revised version of the manuscript).
“Figure 2 and Figure Legend: What is the concentration of yCM and aCM? Was CM depleted of EVs or were EVs + soluble factors determined? Do soluble factors as determined in Figure 2A, contribute to the effects described in Fig. 3-6? Can such important factors functionally bind to EVs (corona) and mediate the anti-aging effects? This possibility should be discussed”.
As indicated in the Material and Methods section, the total conditioned medium (CM) derived from mouse-BMSCs contains both soluble factors and EVs. We wanted to mimic in vitro the aged stromal microenvironment to evaluate its effect on young macrophages, so we think it’s more correct to use the total CM. As we also indicated in the Figure 2 legend, we used 50 µg of yCM and aCM. This information is also added to the Material and Method section 2.4.
The soluble factors determined in figure 2a are derived from mouse-BMSCs, while results presented in figures 3-6 are referred to EVs derived from human-MSCs preconditioned with hypoxia and inflammatory stimuli. For this reason, we don’t think these soluble factors can have an effect on EV functions. On the other hand, a more detailed comparison of HypINF secretomes can be found in our previous paper (doi: 10.1016/j.biomaterials.2020.120633).
“Figure 4 and Results: What concentration of EVs was applied (0.5 µg / xx mL)?”
We used 0.5 mg EVs/well. As indicated in the Materials and Methods section, we plated 2x106 bone marrow-nucleated cells in presence of complete IMDM medium supplemented with M-CSF. After 7 days, macrophages have been cultured in 2 ml complete IMDM medium and treated with 0.5 mg EVs/well.
“The authors should explain the experiments better. As it appears to me, the readout in aged Mφ is 2-3 x stronger than in yMφ, i. e. TNFa: CTRL, EVs, M1, M1+EVs and CD38: CTRL and CTRL+EVs. EVs from MSCs reduce the effect to 50% of the non-treated control, in both, aged and young Mφ. So, the efficacy of EVs might be very similar in both systems, aged versus young. The authors might compare the %-age of change with CTRL as 100%”.
Thanks to the Reviewer for this comment. As required, we have modified the graphs, normalizing the data to the control conditions, as can be seen from the new graphs below. Applying this change, the statistical significances do not change with respect to the corresponding graphs presented in the manuscript. However, we think that it is more appropriate not to modify the graphs of the main figure as we are also interested in showing the differences between controls, i.e. differences in terms of expression of CD38 and TNF-a, already in basal conditions. If we modify the graphs this information could be lost.
Figure 1: (a) Original figure of the article, comparison of the CD38-positive cells from young- and aged- macrophages maintained in CTRL condition and treated or not with 0.5 µg EVs for 24 h. Bars indicate mean ± SD from three independent experiments. * p<0.05, *** p<0.001 (Two-way ANOVA). (b) Percentage of CD38- positive cells in single graphs. * p<0.05. (One-way ANOVA).
Figure 2: (a) Original figure of article, comparison of young- and aged- macrophage’ TNF-α concentrations measured by ELISA in culture supernatant from CTRL, CTRL+EVs, M1 and M1+EVs. Bars indicate mean ± SD from six independent experiments. * p<0.05, ** p<0.01, *** p<0.001, ****p<0.0001 (Two-way ANOVA). (b) Fold-change of TNF-α levels (percentage) of young and aged macrophages in single graphs. * p<0.05, ** p<0.01, *** p<0.001. (One-way ANOVA).
“Does hypoxia matter and to what extent? Only one part of the experiment was performed under hypoxia (generation of EVs). Is there also an effect with normoxia-borne EVs? The author should discuss this issue”.
The Reviewer is right, we decided to use hypoxia just to stimulate hMSCs, together with inflammatory factors (HypINF MSCs). We have previously demonstrated that the characteristics associated with the soluble factors and EVs released by hMSCs stimulated with inflammatory factors in normoxic conditions (NorINF) were similar (doi: 10.1016/j.biomaterials.2020.120633). In the current manuscript, we wanted to focus to the HypINF condition to mimic a typical injury environment, as literature reports indicate that preconditioning represents an adaptive strategy to improve MSC therapeutic efficacy, preparing cells to survive in hostile conditions and enhancing their regulatory activities (doi: 10.1155/2016/3924858; doi: 10.1186/s13287-019-1224-y).
“Figure 5 and Results: An interesting finding that EVs alone are able to produce energy”.
We agree with the Reviewer that the data on the ability of EVs to produce energy on their own are very interesting. Indeed, literature already reports that exosomes and microvesicles can conduct aerobic metabolism (doi: 10.1096/fj.15-279679; doi: 10.1016/j.jprot.2016.02.001; doi: 10.1586/14789450.2015.1055324). However, it is not yet clear what mechanism underlies this metabolic capacity. Some authors suggest that the OxPhos machinery was transferred to the forming vesicles via the endoplasmic reticulum (doi: 10.1096/fj.15-279679; doi: 10.1016/j.jprot.2016.02.001). Others speculate on the inclusion of whole mitochondria in the vesicles (doi: 10.21037/EXRNA-21-30/COIF; doi: 10.1038/s41419-021-03640-9; doi: 10.1101/2021.04.10.439214), although the compatibility of the size of mitochondria and EVs remains open to debate.

Round 2
Reviewer 2 Report
Gorgun et al 2022
The author improved the manuscript and explained my doubts.
However, two issues remain to be solved.
1. Figure 1 need a better layout. The titles of the subfigures need adjustment.
2. I really recommend the authors to indicate the concentrations in a good scientific manner. Thus, indicate the EV concentration in a good scientific, reproducible way. A well is not a defined volume. There are many different plates with different wells and capacities available. Write 0,5 µg/ 2 mL cultures or better, EVs were added to 2 mL cell cultures (0,25 µg/mL) and NOT 0,5 µg /well (also Figure 4, legend).
Author Response
Thanks to the Reviewer for these additional comments that will improve the quality of the manuscript.
- We realized that during the uploading of Figure 1 within the main text, something went wrong. We changed the figure and now the subtitles have been adjusted.
- The Reviewer is completely right. We modified the EV concentration throughout the manuscript, as suggested (and in particular in the "Materials and Methods" section, paragraph "2.8 Macrophage polarization and treatment with EVs" (pg. 6); in the "Results" section, paragraph "3.4 EVs released by Hyp-INF-hMSCs exert an anti-inflammatory role on aged macrophages" (pg. 14); in the "Discussion" section (pg. 21); in the legends associated with figure 4, 5, and 6.
